# Early Injection Laryngoplasty: Acoustic and Aerodynamic Outcomes with a Modified General Anesthesia Approach

**DOI:** 10.3390/medicina61122140

**Published:** 2025-11-30

**Authors:** Esma Altan, Elife Barmak, Dilara Söylemez, Emel Çadallı Tatar

**Affiliations:** 1Ankara Etlik City Hospital, 06170 Ankara, Türkiye; dlrsylmz@gmail.com; 2Department of Speech and Language, Ankara Yıldırım Beyazıt University, 06010 Ankara, Türkiye; elifebarmak@gmail.com; 3Independent Researcher, 06190 Ankara, Türkiye; ectatar@gmail.com

**Keywords:** acoustic analysis, glottic insufficiency, hyaluronic acid, injection laryngoplasty, unilateral vocal fold paralysis, videolaryngostroboscopy

## Abstract

*Background and Objectives*: This study aimed to evaluate postoperative changes in voice quality and glottic function following early injection laryngoplasty with hyaluronic acid performed using a modified general anesthesia approach without airway instrumentation in patients with unilateral vocal fold paralysis. *Materials and Methods*: Thirty-two patients (19 females, 13 males; mean age 51.8 years, range 21–70) who underwent injection laryngoplasty within the first three months after the onset of paralysis were included in this retrospective study. All procedures were performed under general anesthesia without endotracheal intubation, using endoscopic visualization. Objective acoustic and aerodynamic analyses and videostroboscopic examinations were performed preoperatively and postoperatively. Data were analyzed using the Wilcoxon signed-rank test, with *p* < 0.05 considered statistically significant. *Results*: Significant postoperative improvement was observed in acoustic and aerodynamic parameters. Shimmer, jitter, and noise-to-harmonic ratio (NHR) values significantly decreased (*p* < 0.001, *p* < 0.001, and *p* = 0.001, respectively), while maximum phonation time (MFT) increased markedly (*p* < 0.001) and the S/Z ratio decreased (*p* = 0.006). The mean fundamental frequency (F0) decreased slightly but not significantly (*p* = 0.085). Videostroboscopic findings demonstrated improved glottic closure and vocal fold vibration. No major complications occurred. *Conclusions*: Early injection laryngoplasty with hyaluronic acid performed under general anesthesia and endoscopic guidance provides significant improvement in objective voice parameters and glottic efficiency in unilateral vocal fold paralysis. Early intervention appears to enhance phonatory stability and may prevent maladaptive laryngeal changes.

## 1. Introduction

Vocal fold paralysis refers to complete immobility of the vocal fold resulting from neurological injury, whereas vocal fold paresis denotes partial movement impairment [1]. Either condition may affect the larynx unilaterally or bilaterally. These disorders develop due to damage to the vagus nerve along its course from the medulla to the larynx [2]. Etiologies include iatrogenic injuries during thyroid, carotid endarterectomy, anterior cervical spine, skull base, and thoracic surgeries; endotracheal intubation; viral infections; neoplastic causes such as thyroid, lung, esophageal, and mediastinal tumors; neurogenic disorders including amyotrophic lateral sclerosis and cerebrovascular events; and aortic aneurysm, pulmonary artery enlargement, and idiopathic causes [1,3]. Nerve injury may range from transient neuropraxia to complete degeneration, and the resulting glottic configuration significantly influences disease severity [4]. Vocal fold paralysis can be classified based on lesion site, affected nerve, vocal fold position, and the presence of an organic lesion [5].

Symptoms vary depending on whether the paralysis is unilateral or bilateral [2]. In unilateral vocal fold paralysis (UVFP), dysphonia is typically the main presenting symptom. Voice quality may be breathy or hoarse, with reduced pitch and loudness range, limited maximum phonation time, and sometimes diplophonia. Compensatory hyperfunctional behaviors such as false vocal fold compression may lead to strained, harsh, or low-pitched phonation [4]. Iatrogenic causes—particularly thyroidectomy—have become increasingly common [6].

Evaluation of UVFP includes anamnesis, videolaryngoscopy, videolaryngostroboscopy, acoustic and aerodynamic analysis, auditory–perceptual assessment, imaging, and laryngeal electromyography [2,7]. Management options include voice therapy [8], injection laryngoplasty (IL) [9], medialization thyroplasty [10], arytenoid adduction [11], laryngeal reinnervation [12], or combined approaches.

Multiple interventions aim to improve glottic closure and voice quality. Voice therapy is an effective conservative method [13]. IL, first introduced by Bruening in 1911 [14], is widely used to augment the paralyzed fold [15,16]. Materials include autologous fat [17], polydimethylsiloxane [18], hyaluronic acid [19], collagen, calcium hydroxylapatite, and other synthetic injectables [20,21]. Autologous fat injection is an established alternative but requires overcorrection due to early resorption [22]. Advances in instrumentation allow several endoscopic and transcutaneous approaches depending on patient tolerance and surgeon preference.

IL may also be performed under general anesthesia without endotracheal intubation, which provides better visualization during short procedures [23]. High-flow nasal oxygen (HFNO) has been proposed for upper aerodigestive surgery [24,25], while tubeless techniques such as jet ventilation have long been used to maintain an unobstructed surgical field [26]. However, these methods require specialized equipment, may increase procedural complexity, and are not always necessary for very brief interventions that allow for careful physiological monitoring. In selected cases, avoiding airway instrumentation altogether can offer the advantage of an undistorted laryngeal view and a completely unobstructed operative field. Despite this potential, detailed reports describing injection laryngoplasty performed without any airway-supportive technique remain scarce [23], and data on perioperative monitoring strategies and postoperative outcomes are limited. Our study aims to address this gap by presenting early acoustic and aerodynamic results of hyaluronic acid injection laryngoplasty performed under a modified general anesthesia protocol without airway instrumentation.

Medialization thyroplasty (MT) is considered a long-term solution for UVFP and involves creating a cartilage window to insert medializing implants such as silicone, Gore-Tex, or titanium [27,28,29]. MT is often performed under local anesthesia with real-time auditory feedback and may be combined with arytenoid adduction to optimize posterior glottic closure [30].

Although IL has been widely studied, evidence regarding short- and mid-term functional outcomes of hyaluronic acid (HA) remains heterogeneous. HA is preferred due to its biocompatibility, ease of injection, and viscoelastic similarity to the superficial lamina propria [19]. However, data on early-period HA injection and its impact on objective voice outcomes remain limited.

This study therefore aimed to evaluate early hyaluronic acid injection laryngoplasty performed under a modified general anesthesia approach without airway instrumentation in patients with UVFP. Objective acoustic, aerodynamic, and videostroboscopic parameters were compared before and after surgery. We hypothesized that early HA injection performed under unobstructed visualization would result in significant improvements in voice quality and glottic closure.

## 2. Materials and Methods

The study was approved by the Clinical Research Ethics Committee of Etlik City Hospital (Date: 27 September 2023, Decision No: 2023-583). Written informed consent was obtained from all participants prior to inclusion.

### 2.1. Study Design and Participants

This retrospective observational study adhered to the STROBE guidelines. A total of 55 consecutive patients who underwent injection laryngoplasty for the treatment of unilateral vocal fold paralysis (UVFP) between January 2023 and September 2025 at the Etlik City Hospital Laryngology Clinic were evaluated. Of the 55 patients evaluated, 32 met the inclusion criteria and had complete pre- and postoperative data, and were therefore included in the final analysis. The diagnosis of UVFP was established based on videolaryngoscopic findings and clinical history. As this was a retrospective study, the sample size was determined by the total number of eligible patients treated during the study period; therefore, no a priori power calculation was performed.

Patient selection followed predefined inclusion and exclusion criteria. All adult patients with a confirmed diagnosis of UVFP and complete pre- and postoperative evaluations were included. The etiological distribution comprised idiopathic, iatrogenic, and malignancy-related causes.

Patients were excluded if they had: (1) bilateral vocal fold paralysis; (2) a history of laryngeal framework surgery—including prior medialization thyroplasty, arytenoid adduction, or previous injection laryngoplasty; (3) malignant laryngeal lesions; (4) contralateral vocal fold pathology such as polyps, cysts, sulcus vocalis, or scarring; (5) significant structural laryngeal abnormalities; or (6) incomplete clinical or follow-up data. These clarifications were made to ensure transparent and reproducible patient selection.

Missing data were managed using a complete-case analysis approach, and no patients were lost during the 1-month postoperative follow-up period. Demographic information, etiology of paralysis, and pre- and postoperative acoustic, aerodynamic, and videostroboscopic outcomes were recorded for all participants.

All videostroboscopic examinations were performed by the same laryngologist using a rigid 70° endoscope under constant light and stroboscopic illumination (XION GmbH, Berlin, Germany). Acoustic and aerodynamic measurements were obtained using standardized protocols to ensure consistency across evaluations. (Voice recordings were obtained using the KayPENTAX Model 4500 CSL^®^ (Computerized Speech Lab) system (KayPENTAX, Lincoln Park, NJ, USA) and a dynamic microphone (Shure SM48-LC; Shure Inc., Chicago, IL, USA), positioned approximately 10 cm from the mouth at a 45° angle. Acoustic parameters were analyzed using the Multidimensional Voice Program (MDVP, version 7.1; KayPENTAX, Lincoln Park, NJ, USA).

No postoperative voice therapy was initiated during the first month following injection laryngoplasty, as the aim of the study was to evaluate the isolated early physiologic effect of the procedure without the confounding influence of rehabilitation.

### 2.2. Anesthetic Technique

All injection laryngoplasty procedures were performed in the operating room under general anesthesia without endotracheal intubation or additional airway instrumentation. Anesthesia was induced and maintained with propofol and remifentanil infusions. Neuromuscular blockade with rocuronium was administered only when minimal vocal fold movement was observed, based on the anesthesiologist’s discretion.

All patients underwent pre-oxygenation with 100% oxygen via face mask, after which oxygen saturation and physiological parameters were continuously monitored, including ECG, non-invasive blood pressure, and pulse oximetry. End-tidal CO_2_ was observed via mask sampling when feasible. Adequate oxygenation was maintained throughout the procedure, consistent with previous reports using similar short-duration techniques [23,31,32,33]. No desaturation events (<92%) occurred in any patient.

Because the intervention time for each case was only a few minutes, the anesthesia team predefined safety limits, including termination or conversion to standard ventilation if oxygen saturation dropped below 92% or if laryngeal exposure was insufficient.

For airway exposure, the anesthesiologist positioned a standard operating laryngoscope, and the surgeon performed the injection under direct visualization with a 0-degree rigid endoscope. This approach provided a completely unobstructed laryngeal view, facilitated precise injection placement, and minimized potential risks associated with endotracheal intubation or jet ventilation. All patients were discharged on the same day after a brief postoperative observation period.

### 2.3. Surgical Technique

Injection laryngoplasty was performed within the first three months following the onset of paralysis in all patients. After adequate exposure of the glottis was achieved, the paralyzed vocal fold was injected with hyaluronic acid (Dexell^®^, 1 mL prefilled syringe; HİJYENİK TIBBİ ÜRÜNLER SAN. ve TİC. A.Ş., Istanbul, Türkiye) using a 1 mL syringe and a 25-gauge long injection needle. The injection was made into the lateral aspect of the vocal fold, slightly lateral to the vocal process, until optimal medialization was observed. Minor overcorrection was intentionally performed to compensate for early postoperative resorption. Proper glottic closure and symmetry were confirmed endoscopically before completion of the procedure.

### 2.4. Voice Evaluation

Voice assessments were performed preoperatively and at one month postoperatively.

### 2.5. Objective Acoustic Evaluation

Voice recordings were obtained in a sound-treated room with the patient in a seated position. The subjects were instructed to sustain the vowel /*a*/ at a comfortable pitch and intensity for at least three seconds. Acoustic parameters including fundamental frequency (F0), jitter (%), shimmer (%), and noise-to-harmonic ratio (NHR) were analyzed. All recordings were made with the same microphone-to-mouth distance and under identical environmental conditions to ensure reproducibility.

### 2.6. Statistical Analysis

Statistical analyses were performed using IBM SPSS Statistics for Windows, Version 22.0 (IBM Corp., Armonk, NY, USA). Descriptive statistics were calculated for all variables and presented as mean ± standard deviation, minimum, and maximum values. Data distribution was assessed using the Shapiro–Wilk test. Since the acoustic and aerodynamic parameters did not follow a normal distribution, non-parametric tests were applied. Preoperative and postoperative measurements were compared using the Wilcoxon signed-rank test. No formal correction for multiple comparisons was performed due to the exploratory nature of the study and the limited sample size; this has been acknowledged as a study limitation. A *p*-value of <0.05 was considered statistically significant.

## 3. Results

Thirty-two patients (19 females, 13 males) who underwent injection laryngoplasty for unilateral vocal fold paralysis were included in the study. The patients’ ages ranged from 21 to 70 years (mean = 51.84, median = 54.00). The left vocal fold was affected in 59.4% the right one is 40.6% of the patients, and the most common etiology was iatrogenic (65.6%), followed by idiopathic (21.9%) and malignant (12.5%) causes (Table 1).

The descriptive analysis (Table 2) revealed that most acoustic parameters improved markedly after treatment. The mean fundamental frequency (F0) decreased slightly from 217.5 Hz preoperatively to 200.8 Hz postoperatively, while measures reflecting perturbation in the voice signal showed clear improvement: shimmer decreased from 6.69 to 2.73 and jitter from 11.72 to 7.06. The noise-to-harmonic ratio (NHR) was also reduced (from 0.33 to 0.17), indicating a cleaner and more stable acoustic signal. Maximum phonation time (MFT) increased substantially, from 4.25 to 9.00 s, and the mean S/Z ratio improved from 2.06 to 1.34, reflecting enhanced glottic efficiency and respiratory control.

The Wilcoxon signed-rank test confirmed that these improvements were statistically significant for shimmer (*p* < 0.001), jitter (*p* < 0.001), NHR (*p* = 0.001), MFT (*p* < 0.001), and S/Z ratio (*p* = 0.006) (Table 3). The change in F0 did not reach statistical significance (*p* = 0.085). Overall, both descriptive and inferential analyses demonstrated a consistent postoperative enhancement in voice stability and phonatory efficiency following hyaluronic acid injection.

These findings confirm that injection laryngoplasty with hyaluronic acid is an effective and well-tolerated procedure for improving voice quality, glottic efficiency, and overall phonatory function in patients with unilateral vocal fold paralysis. No intraoperative complications occurred.

## 4. Discussion

Phonation begins with the adduction of the vocal folds, and intact abduction–adduction movements are essential for normal voice production and control [34,35]. In cases of glottal insufficiency, incomplete closure of the vocal folds leads to excessive air leakage and a characteristic breathy voice quality [36]. Therefore, one of the main therapeutic goals in the management of unilateral vocal fold paralysis is to reduce the glottal gap and enhance glottic closure. Previous studies have suggested that performing injection laryngoplasty early after the onset of paralysis may preserve tactile feedback between the vocal folds and promote more favorable glottic positioning as synkinetic reinnervation occurs [37]. Early intervention may also prevent or minimize laryngeal muscle atrophy, although the relationship between the timing of paralysis and the durability of treatment effects remains uncertain due to non-randomized designs and heterogeneous follow-up periods in prior research.

In the present study, all injections were performed within the first three months of paralysis under general anesthesia using endoscopic guidance and hyaluronic acid as the injectable material. This early intervention strategy resulted in satisfactory medialization and significant improvements in voice quality, supporting the notion that early injection laryngoplasty may enhance glottic compensation and prevent maladaptive muscular changes. These findings align with previous reports demonstrating that early medialization contributes to better phonatory stability and overall vocal recovery. Given the possibility of spontaneous recovery in unilateral vocal fold paralysis, the use of a temporary injectable such as hyaluronic acid is particularly appropriate.

Injection laryngoplasty continues to evolve with respect to materials, approaches, and techniques. Hyaluronic acid has become one of the most widely used injectables due to its favorable viscoelastic properties, its ability to maintain vocal fold pliability, and its negligible risk of immunogenic reaction [38,39,40]. As a temporary material with gradual resorption, HA provides reliable short-term medialization [20]. Consistent with previous studies demonstrating postoperative improvement in perturbation-based acoustic parameters following HA injection [41,42], our results confirm significant gains in acoustic, aerodynamic, and stroboscopic findings through multidimensional evaluation using MDVP-based analysis.

Numerous reports have described both transoral and percutaneous injection approaches with favorable outcomes and low complication rates [43,44,45,46]. Examples include studies by Lee et al. [47], Chandran et al. [48], and Mohammed et al. [49], all showing significant improvements in perceptual measures and VHI-10 scores. Other authors have similarly demonstrated enhanced Voice Performance Questionnaire (VPQ) scores and perceptual parameters [50]. In contrast to these studies, the present work focused exclusively on objective acoustic and aerodynamic measures rather than perceptual or patient-reported outcomes. While this allowed for precise quantification of physiologic changes, subjective assessments would provide additional insight into patient experience and could be incorporated in future research.

The present cohort demonstrated significant postoperative decreases in shimmer, jitter, and noise-to-harmonic ratio (NHR), along with marked improvement in maximum phonation time (MFT) and S/Z ratio, reflecting enhanced glottic efficiency. These findings are consistent with previous studies utilizing the Multi-Dimensional Voice Program (MDVP) and aerodynamic metrics to evaluate treatment success in voice disorders [51,52]. Together, these objective improvements strongly support the effectiveness of hyaluronic acid injection in restoring balanced and stable phonatory function.

Regarding material durability, hyaluronic acid provides reliable early-phase glottic compensation but has a shorter duration of effect compared with longer-acting injectables such as calcium hydroxylapatite (CaHA) [53] or autologous fat [54,55]. CaHA often offers long-term augmentation, while autologous fat provides biocompatibility with variable resorption behavior. Patients requiring continued medialization may benefit from repeat injections, medialization thyroplasty, or more durable injectables depending on symptom recurrence and glottic configuration. Additionally, although objective measures were the focus of the present analysis, perceptual ratings and patient-reported outcomes such as VHI-10 and GRBAS are increasingly recognized as essential tools for a comprehensive assessment of clinical effectiveness [55].

Although videostroboscopy was performed for all patients, detailed semi-quantitative parameters—such as glottic closure pattern, vibratory amplitude, mucosal wave, and phase symmetry—were not recorded systematically. Nevertheless, the robust improvements in acoustic parameters suggest favorable vibratory changes following medialization. Future work incorporating validated stroboscopic scoring systems and standardized image archiving would allow a clearer correlation between vibratory characteristics and functional outcomes.

An additional contribution of this study is the use of a tubeless endolaryngeal approach. While tubeless techniques have long been applied in selected airway procedures, detailed descriptions of injection laryngoplasty performed without any airway-supportive methods remain limited in the literature. In our cohort, the modified general anesthesia protocol—characterized by preoxygenation, continuous physiologic monitoring, and strictly time-restricted instrumentation—enabled the procedure to be completed safely without desaturation or airway instability. This method provided an entirely unobstructed view of the glottis, avoided tube-related distortion, and facilitated precise, controlled needle placement. Optimal visualization of the vocal folds allowed the injectable material to be delivered accurately into the intended plane, enabling a clear assessment of the physiologic effects during the early period in which hyaluronic acid has not yet undergone significant resorption. By documenting perioperative safety observations together with early acoustic and aerodynamic outcomes obtained under these conditions, the present study contributes physiologic data that help clarify the feasibility and safety considerations of performing injection laryngoplasty without airway instrumentation, an area that remains underreported in current literature

Although recent research emphasizes the advantages of local-anesthesia injections—such as real-time voice feedback and reduced cost—our findings highlight the distinct value of tubeless general-anesthesia injections. Unobstructed visualization and stable procedural conditions allow accurate medialization and consistent assessment of vocal fold behavior. By providing multidimensional objective data in the early period after paralysis onset, this study contributes insight that complements, rather than duplicates, existing literature on office-based injection laryngoplasty.

Finally, this work represents the initial step in a planned series of investigations. Future studies will directly compare the tubeless general-anesthesia approach with office-based local-anesthesia injection laryngoplasty to determine whether both techniques offer comparable precision, patient tolerance, and early acoustic and aerodynamic outcomes. Such comparative analyses will help clarify the relative advantages and limitations of each method and expand the evidence base for optimal management of unilateral vocal fold paralysis.

### Limitations

This study has several limitations. First, the sample size was relatively small, and an a priori power calculation could not be performed due to the retrospective design; therefore, the statistical power of the findings may be limited. Second, only short-term postoperative outcomes were assessed. Because hyaluronic acid is a resorbable material, medium- and long-term results may differ as resorption progresses, and longer follow-up is needed to evaluate the durability of the treatment effect.

Third, as with all retrospective studies, data collection was subject to potential issues such as incomplete documentation, variability in record quality, and possible selection bias. In addition, no formal correction for multiple comparisons was applied. Given the exploratory nature of the study and the limited sample size, applying strict adjustments would have markedly increased the risk of type II error.

Perceptual assessments (such as GRBAS or VHI-10) and patient-reported outcomes were not available for all cases in the medical records and therefore could not be included. This limited our ability to correlate physiologic improvement with subjective voice outcomes.

Another limitation relates to the classification of iatrogenic cases. Because surgical documentation did not consistently specify the mechanism of nerve injury, it was not possible to distinguish whether recurrent laryngeal nerve dysfunction resulted from direct trauma, traction, thermal injury, or necessary oncologic dissection. For this reason, all surgery-associated cases were categorized as iatrogenic, consistent with current literature.

Videostroboscopy was performed in all patients; however, formal semi-quantitative stroboscopic scoring (e.g., amplitude, symmetry, mucosal wave) was not routinely documented, and high-quality still images were not consistently archived. As a result, detailed vibratory analysis and illustrative figures could not be included in this report.

Additionally, because most procedures were performed within a narrow postoperative window (<3 months), meaningful stratification by timing subgroups was not possible.

Postoperative care was standardized across all patients and consisted of routine voice rest recommendations and monitoring; however, detailed postoperative adherence data were not consistently documented and could not be analyzed.

Lastly, the study focused specifically on early postoperative outcomes following a tubeless endolaryngeal approach, without endotracheal intubation. While this design allowed evaluation of immediate physiologic changes, longer-term studies comparing this technique with office-based injection laryngoplasty under local anesthesia are needed to determine relative advantages, durability, and patient-centered outcomes.

Future prospective studies with larger cohorts, standardized stroboscopic scoring, systematic image archiving, longer follow-up intervals, and comparative designs will be essential to validate and expand upon these findings.

## 5. Conclusions

Injection laryngoplasty with hyaluronic acid performed under general anesthesia without airway instrumentation and with direct endoscopic guidance resulted in significant improvements in objective acoustic and aerodynamic parameters in patients with unilateral vocal fold paralysis. The procedure provided enhanced vocal stability, improved glottic efficiency, and satisfactory medialization, with no intraoperative complications observed under the carefully monitored conditions used in this study. These findings support the value of early hyaluronic acid injection as a safe and effective option for restoring phonatory function. Further prospective studies incorporating perceptual measures, standardized stroboscopic scoring, and longer-term follow-up will help clarify the relative advantages and broader clinical applicability of this anesthesia approach in injection laryngoplasty.

## Figures and Tables

**Table 1 medicina-61-02140-t001:** Demographic and Clinical Characteristics of the Study Population.

		Frequency	Percent	Valid Percent	Cumulative Percent
Valid/SEX	Male	13	40.6	40.6	40.6
	Female	19	59.4	59.4	100.0
	Total	32	100.0	100.0	
Valid/SIDE	Right	13	40.6	40.6	40.6
	Left	19	59.4	59.4	100.0
	Total	32	100.0	100.0	
Valid/ETIOLOGY	Idiopathic	7	21.9	21.9	21.9
	Iatrogenic	21	65.6	65.6	87.5
	Malignancy	4	12.5	12.5	100.0
	Total	32	100.0	100.0	

(Sex, side of paralysis, and etiology distribution among 32 patients).

**Table 2 medicina-61-02140-t002:** Descriptive Statistics of Acoustic and Aerodynamic Parameters Before and After Injection Laryngoplasty.

	N	Mean	Std. Deviation	Minimum	Maximum	Percentiles		
						25th	50th (Median)	75th
F0_Before inj	32	217.49963	58.533108	71.700	321.203	178.63375	220.71850	262.07000
Shimmer_Before inj	32	6.69303	4.947742	0.494	20.500	3.60625	5.28550	9.68925
Jitter_Before inj	32	11.71681	7.620636	3.407	34.247	6.59675	9.32000	13.68425
NHR_Before inj	32	0.32500	0.240240	0.085	1.026	0.14800	0.20350	0.47400
MFT Before inj	32	4.25	4.399	1	26	2.25	3.00	5.00
S/Z Ratio Before inj	32	2.0578	1.38919	0.70	6.00	1.1000	1.7000	2.7250
F0 After inj	32	200.80163	67.054363	100.816	343.690	146.60150	193.73250	236.42125
Shimmer After inj	32	2.72572	2.515304	0.465	10.136	1.28250	1.71200	3.05525
Jitter_After inj	32	7.058172	6.2450463	2.5720	29.1680	3.975250	5.010500	7.435750
NHR_After inj	32	0.16963	0.101085	0.094	0.547	0.11575	0.12850	0.16075
MFT After inj	32	9.00	4.288	1	18	6.00	8.50	11.75
S/Z Ratio After inj	32	1.3375	0.48160	0.90	3.00	1.0000	1.1000	1.5750

(Mean, standard deviation, minimum, maximum, and percentiles of F0, jitter, shimmer, NHR, MPT, and S/Z ratio). Abbreviations: F0—Fundamental Frequency; NHR—Noise-to-Harmonic Ratio; MPT—Maximum Phonation Time; S/Z—Ratio of sustained/s/and/z/phonemes; Before-inj—Before Injection; After-inj—After Injection.

**Table 3 medicina-61-02140-t003:** Comparison of Acoustic and Aerodynamic Parameters Before and After Injection Laryngoplasty (Wilcoxon Signed-Rank Test).

	F0 After Inj–F0 Before Inj	Shimmer After Inj–Shimmer Before Inj	Jitter After Inj–Jitter Before Inj	NHR After Inj–NHR Before Inj	MFT After Inj–MFT Before Inj	S/Z Ratio After Inj–S/Z Ratio Before Inj
Z	−1.720 ^b^	−3.702 ^b^	−3.646 ^b^	−3.441 ^b^	−4.298 ^c^	−2.755 ^b^
Asymp. Sig. (2-tailed)	0.085	0.000	0.000	0.001	0.000	0.006

^b^ Based on positive ranks. ^c^ Based on negative ranks. (Statistical comparison of pre- and postoperative measurements following hyaluronic acid injection laryngoplasty). Abbreviations: F0—Fundamental Frequency; NHR—Noise-to-Harmonic Ratio; MPT—Maximum Phonation Time; S/Z—Ratio of sustained/s/and/z/phonemes; Before-inj—Before Injection; After-inj—After Injection.

## Data Availability

The data presented in this study are available on request from the corresponding author. The data are not publicly available due to institutional privacy regulations and ethical restrictions related to patient confidentiality.

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
