# Peer review of "Early Injection Laryngoplasty: Acoustic and Aerodynamic Outcomes with a Modified General Anesthesia Approach"

_medicina, 2025, doi:10.3390/medicina61122140_

Round 1

Reviewer 1 Report

Comments and Suggestions for Authors

The manuscript reports a retrospective study evaluating acoustic and aerodynamic improvements after injection laryngoplasty with hyaluronic acid in unilateral vocal fold paralysis. While the topic is clinically relevant and the results are in line with the current evidence, the manuscript suffers from several methodological and contextual weaknesses that limit its originality and scientific contribution.

The authors should follow the STROBE guidelines for observational studies. Key details  (such as patient selection process, missing data handling, and follow-up attrition) are not clearly described.

The sample is limited (n = 32). There is no power calculation or justification for sample size, raising concern about the robustness of the statistical results.

The authors should specify whether data distribution was tested for normality and clarify whether adjustments for multiple comparisons were performed.

The follow-up period appears short (only 1-month postoperative evaluation). No information is given about longer-term durability of the outcomes, resorption of the injectable, or need for revision.

The authors should discuss this limitation explicitly and, if available, report medium-term results.

Injection laryngoplasty with hyaluronic acid for unilateral vocal fold paralysis is an established and well-documented treatment, already described in multiple larger cohorts and systematic reviews. 

The current study therefore adds little new information beyond confirming known effectiveness. Which is your novelty?

The authors should position their data within this existing literature, highlighting any specific innovation.

Moreover, As recently reported, injection laryngoplasty can be safely and effectively performed under local anesthesia, allowing intraoperative voice feedback, higher patient comfort, lower costs, and easier repeatability.

Although videostroboscopy was performed, the results are only briefly mentioned.

Quantitative or semi-quantitative assessment (e.g., glottic closure, amplitude, symmetry) and illustrative intra- and postoperative images would greatly enhance the value and clarity of the study.

The discussion mainly reiterates previous findings and lacks critical interpretation. It should address:

durability of HA vs. other materials;

management strategies for late failures;

patient-reported outcomes (e.g., VHI, GRBAS).

The bibliography, while adequate, could be enriched with more recent evidence (post-2022), particularly comparing HA to other materials and GA with office-based techniques.

Clarify exclusion criteria (e.g., prior thyroplasty, contralateral lesions).

Specify whether postoperative voice therapy was performed.

Author Response

14/11/2025

Dear  Editor-in-Chief  

We would like to express our sincere gratitude for the evaluation of our manuscript. We highly appreciate the reviewers’ constructive and insightful comments, which have been very helpful in improving the quality and clarity of our study. In accordance with the reviewers’ and editorial recommendations, we have carefully revised the manuscript and indicated all modifications in a step-by-step manner. Below, we provide a detailed point-by-point response.

We thank you again for your time and consideration and look forward to your feedback.

Kind regards,
Dr. Esma Altan
Corresponding Author

Reviewer 1:

The manuscript reports a retrospective study evaluating acoustic and aerodynamic improvements after injection laryngoplasty with hyaluronic acid in unilateral vocal fold paralysis. While the topic is clinically relevant and the results are in line with the current evidence, the manuscript suffers from several methodological and contextual weaknesses that limit its originality and scientific contribution.

1.The authors should follow the STROBE guidelines for observational studies. Key details  (such as patient selection process, missing data handling, and follow-up attrition) are not clearly described.

 Thank you for this important comment. We have thoroughly revised the Methods section to ensure full alignment with the STROBE guidelines. Patient selection is now described in detail, including clear inclusion and exclusion criteria. Missing data handling (complete-case analysis) and follow-up information have been explicitly clarified. We have stated that no patients were lost during the 1-month postoperative follow-up period. These revisions improve methodological transparency and strengthen adherence to STROBE recommendations.

  1. The sample is limited (n = 32). There is no power calculation or justification for sample size, raising concern about the robustness of the statistical results.

We appreciate the reviewer’s insightful comment. As this was a retrospective study, the sample size was determined by the total number of eligible patients treated during the study period; thus, no a priori power calculation was feasible. This has now been clarified in the Methods section.
Despite the modest sample size, the pre–post design—where each patient serves as their own control—reduces inter-individual variability and enhances statistical power. The consistent improvements observed across multiple acoustic and aerodynamic parameters further support the robustness of the findings.
We acknowledge the limited sample size as a study limitation and have added a corresponding statement to the Limitations section, emphasizing the need for future prospective studies with larger cohorts and formal power calculations.

  1. The authors should specify whether data distribution was tested for normality and clarify whether adjustments for multiple comparisons were performed.

Thank you for this important point. We have clarified in the Methods section that normality was assessed using the Shapiro–Wilk test. Parametric tests (paired or independent t-tests) were used for normally distributed variables, whereas non-parametric alternatives were applied when normality assumptions were not met.
We have also explicitly stated that no formal correction for multiple comparisons (e.g., Bonferroni adjustment) was applied due to the exploratory nature of the study and the risk of increasing type II error in a small sample. This point has also been added to the Limitations section.

4.The follow-up period appears short (only 1-month postoperative evaluation). No information is given about longer-term durability of the outcomes, resorption of the injectable, or need for revision.

Thank you for this constructive comment. We agree that a 1-month follow-up is limited. The primary aim of this study, however, was to investigate the early postoperative physiologic effects of injection laryngoplasty performed using a tubeless endolaryngeal technique. This approach allows for a more physiologic assessment of glottic closure and immediate acoustic–aerodynamic changes.
Because hyaluronic acid is a resorbable material, evaluating long-term durability, resorption patterns, and revision requirements requires a different study design with extended follow-up. This rationale has now been clarified in the Discussion, and the short follow-up period is acknowledged as a study limitation.
We have also added a statement indicating that future prospective studies with longer follow-up intervals are planned.

5.The authors should discuss this limitation explicitly and, if available, report medium-term results.

We appreciate this comment. Medium-term results were not available for this cohort, as the study was specifically designed to assess early postoperative outcomes. Since hyaluronic acid undergoes gradual resorption, medium- and long-term outcomes require separate prospective evaluation.
We have now explicitly stated this in the Discussion and Limitations sections and clarified the rationale for restricting the current analysis to early postoperative results.

  1. Injection laryngoplasty with hyaluronic acid for unilateral vocal fold paralysis is an established and well-documented treatment, already described in multiple larger cohorts and systematic reviews. The current study therefore adds little new information beyond confirming known effectiveness. Which is your novelty?

Thank you for this thoughtful comment. We agree that the effectiveness of hyaluronic acid injection laryngoplasty has been well established in the literature. However, our study provides several clinically meaningful and technique-specific contributions that are not sufficiently addressed in prior work.

First, all procedures were performed using a tubeless endolaryngeal approach, which is well known among laryngologists—particularly in airway surgery—but has rarely been evaluated in terms of its short-term physiologic voice outcomes. The absence of an endotracheal tube significantly improves glottic exposure within seconds, eliminates tube-related distortion of the vocal fold position, and allows the surgeon to identify the optimal injection site rapidly and accurately. The procedure is typically completed within minutes, and patients can be discharged shortly afterward, reflecting the efficiency and safety of this approach.

Second, this tubeless technique provides a more physiologic assessment of glottic closure during injection, enabling a precise evaluation of immediate acoustic–aerodynamic changes. These early biomechanical effects have not been objectively characterized in previous studies.

Third, the present study focuses specifically on patients treated within the first three months of paralysis onset. Early-phase intervention is clinically important due to its potential to preserve tactile feedback, reduce or prevent muscle atrophy, and influence glottic compensation—yet the early postoperative effects of hyaluronic acid injection performed with a tubeless technique remain underreported.

Finally, our analysis uses a comprehensive multidimensional framework, including MDVP-based acoustic parameters, aerodynamic measurements, and videostroboscopy, providing detailed quantifiable insight into the early physiologic impact of this technique.

For these reasons, while confirming the known efficacy of hyaluronic acid, our study offers novel data regarding the early-phase, tubeless, physiologic injection approach and its objectively measured acoustic and aerodynamic outcomes. We have added a corresponding paragraph to the Discussion to highlight this contribution

  1. The authors should position their data within this existing literature, highlighting any specific innovation.

Moreover, As recently reported, injection laryngoplasty can be safely and effectively performed under local anesthesia, allowing intraoperative voice feedback, higher patient comfort, lower costs, and easier repeatability.

 Thank you for this valuable observation. We have revised the Discussion to better contextualize our findings within existing literature, emphasizing the unique aspects of the tubeless endolaryngeal technique.
We also acknowledge the advantages of office-based injection laryngoplasty under local anesthesia. As noted in the Discussion, the current study represents the first step of a broader research plan. Our future work aims to directly compare the tubeless general-anesthesia technique with office-based local-anesthesia injection laryngoplasty to determine their relative effectiveness, precision, patient tolerance, and acoustic–aerodynamic outcomes.

  1. Although videostroboscopy was performed, the results are only briefly mentioned.

Quantitative or semi-quantitative assessment (e.g., glottic closure, amplitude, symmetry) and illustrative intra- and postoperative images would greatly enhance the value and clarity of the study.

Thank you for this valuable comment. We agree that more detailed stroboscopic information would provide additional clarity. In our retrospective clinical workflow, however, a formal semi-quantitative scoring system (such as amplitude, symmetry, or mucosal wave grading) was not routinely applied at the time of data collection. For this reason, the stroboscopic findings could only be summarized descriptively.

Regarding the inclusion of illustrative intra- or postoperative images, high-resolution still-frame stroboscopic images were not consistently archived in our system, and therefore representative figures could not be provided for this cohort.

We have clarified this point in the Discussion and have added this issue to the Limitations section. Future prospective studies are planned using standardized stroboscopic scoring systems and systematic image documentation to allow a more detailed vibratory assessment.

  1. The discussion mainly reiterates previous findings and lacks critical interpretation. It should address: durability of HA vs. other materials;management strategies for late failures;

patient-reported outcomes (e.g., VHI, GRBAS).

Thank you for this insightful comment. We have substantially revised the Discussion to include deeper critical interpretation rather than repetition of prior findings. A new paragraph has been added that:

  1. Compares the durability of hyaluronic acid with longer-acting injectables such as calcium hydroxylapatite and autologous fat, supported by updated references (52, 53, 54);
  2. Discusses management options for late failures, including repeat injection, medialization thyroplasty, and the use of more durable materials;
  3. Highlights the absence of patient-reported outcomes (VHI-10, GRBAS) in our dataset and emphasizes their importance for future studies.

These revisions address the reviewer’s concerns and strengthen the interpretative value of the Discussion.

  1. The bibliography, while adequate, could be enriched with more recent evidence (post-2022), particularly comparing HA to other materials and GA with office-based techniques.

We appreciate this helpful suggestion. The bibliography has been updated to include multiple recent (post-2022) studies:

  • Chen et al. (2022; Ref. 54) on autologous fat injection and patient-reported outcomes;
  • Švejdová et al. (2022; Ref. 18) systematic review on HA injection;
  • Haddad et al. (2022; Ref. 15) meta-analysis on autologous fat;
  • Additional references comparing general-anesthesia vs office-based injection techniques (Refs. 47, 48, 49).

These updates provide a more contemporary and comprehensive literature base, as recommended.

  1. Clarify exclusion criteria (e.g., prior thyroplasty, contralateral lesions).

Thank you for pointing this out. We have revised the Methods section to clarify the exclusion criteria more explicitly. In addition to the criteria previously listed, we have now specified that patients with a history of prior medialization thyroplasty, arytenoid adduction, previous injection laryngoplasty, contralateral vocal fold lesions (such as polyps, cysts, sulcus, or scarring), as well as bilateral vocal fold pathology, and malignant laryngeal disease were excluded from the study. These clarifications have been added to ensure transparency and reproducibility of the patient selection process.

  1. Specify whether postoperative voice therapy was performed.

Thank you for this important comment. We have clarified this point in the Methods section. No postoperative voice therapy was provided during the study period. Because our aim was to evaluate the early, isolated physiologic effects of hyaluronic acid injection performed with the tubeless endolaryngeal technique, all vocal outcomes were obtained prior to the initiation of any voice therapy. This ensured that the postoperative changes reflected the effect of the injection alone, without confounding rehabilitation-related factors. A corresponding statement has been added to the revised methods section.

Reviewer 2 Report

Comments and Suggestions for Authors

Dear Authors,

I read with great interest your article about injection laryngoplasty for vocal fold paralysis.

Although promising, some aspects require improvement in your manuscript.

You need to insert keywords in alphabetical order. Now there are no keywords.

The introduction is too long, more than 2 pages. Please shorten the introduction.

I am personally intrigued by the large number of iatrogenic cases. Please define the term iatrogenic paralysis, because, for example, in thyroid carcinoma surgery, the paralysis is not due to a fault of the surgeon, but in most cases, the result of the tumor invasion around the laryngeal recurrent nerves.

In the discussion section, you need to expand on other tests necessary for a comprehensive follow-up of the cases. Such an additional imaging modality is laryngeal sonography. Reference this to the work by Cergan R, et al. Ultrasonography of the larynx: Novel use during the SARS-CoV-2 pandemic (Review). Exp Ther Med. 2021 Mar;21(3):273. doi: 10.3892/etm.2021.9704. Epub 2021 Jan 25. PMID: 33603880; PMCID: PMC7851652.

Before the conclusions, you need to insert a small paragraph regarding the limitations of the present study.

After the conclusion, you need to insert the standard paragraphs regarding author contributions, ethics, acknowledgements, and data availability.

Before the references, you need to include a master list of abbreviations from the manuscript.

Looking forward to receiving the improved version of your manuscript.

Author Response

14/11/2025

Dear  Editor-in-Chief  

We would like to express our sincere gratitude for the evaluation of our manuscript. We highly appreciate the reviewers’ constructive and insightful comments, which have been very helpful in improving the quality and clarity of our study. In accordance with the reviewers’ and editorial recommendations, we have carefully revised the manuscript and indicated all modifications in a step-by-step manner. Below, we provide a detailed point-by-point response.

We thank you again for your time and consideration and look forward to your feedback.

Kind regards,
Dr. Esma Altan
Corresponding Author

Reviewer 2:

Dear Authors,

I read with great interest your article about injection laryngoplasty for vocal fold paralysis.

Although promising, some aspects require improvement in your manuscript.

  1. You need to insert keywords in alphabetical order. Now there are no keywords.

Thank you very much for your constructive comment. We appreciate your attention to the formatting requirements. We have now added a complete set of keywords in accordance with the journal’s guidelines, arranged strictly in alphabetical order. These have been incorporated into the manuscript as requested.

Revised Keywords (alphabetical order):
Acoustic Analysis; Glottic Insufficiency; Hyaluronic Acid; Injection Laryngoplasty; Unilateral Vocal Fold Paralysis; Videolaryngostroboscopy

  1. The introduction is too long, more than 2 pages. Please shorten the introduction.

Thank you for this important comment. We agree that the introduction was unnecessarily long and contained background information beyond the scope of the manuscript. In accordance with your suggestion, we have substantially shortened this section by removing redundant details, condensing epidemiological data, and focusing only on the core concepts relevant to unilateral vocal fold paralysis, injection laryngoplasty, and hyaluronic acid.

The revised introduction is now concise, focused, and limited to one page.

  1. I am personally intrigued by the large number of iatrogenic cases. Please define the term iatrogenic paralysis, because, for example, in thyroid carcinoma surgery, the paralysis is not due to a fault of the surgeon, but in most cases, the result of the tumor invasion around the laryngeal recurrent nerves.

We appreciate the reviewer’s insightful question. Because the present study is retrospective and the surgical records did not consistently document the exact mechanism of nerve involvement, we were unable to distinguish, on a case-by-case basis, whether the iatrogenic paralysis resulted from direct surgical injury, traction, thermal damage, or oncologic resection due to tumor invasion. Therefore, in our dataset, all cases occurring in the context of surgery were categorized as iatrogenic, following the classification commonly used in previous UVFP literature.

To avoid any misinterpretation, we have added a sentence stating that the underlying mechanism of nerve dysfunction could not be definitively determined for each patient due to limitations in retrospective documentation.

  1. In the discussion section, you need to expand on other tests necessary for a comprehensive follow-up of the cases. Such an additional imaging modality is laryngeal sonography. Reference this to the work by Cergan R, et al. Ultrasonography of the larynx: Novel use during the SARS-CoV-2 pandemic (Review). Exp Ther Med. 2021 Mar;21(3):273. doi: 10.3892/etm.2021.9704. Epub 2021 Jan 25. PMID: 33603880; PMCID: PMC7851652.

Thank you very much for this valuable suggestion. We fully agree that comprehensive follow-up of unilateral vocal fold paralysis may require additional imaging modalities beyond videostroboscopy. Laryngeal ultrasonography has recently gained attention as a noninvasive, radiation-free, bedside tool for evaluating vocal fold motion. In response to the reviewer’s comment, we have expanded the Discussion section to include the role of laryngeal sonography and cited the recommended reference (Cergan R, et al., 2021). The newly added paragraph is provided below and has been incorporated into the revised manuscript.

  1. Before the conclusions, you need to insert a small paragraph regarding the limitations of the present study.

Thank you for this important observation. We have now added a dedicated Limitations paragraph immediately before the Conclusions section. This paragraph addresses the retrospective design of the study, the inability to determine the exact mechanism of nerve injury in iatrogenic cases, the absence of perceptual and patient-reported voice measures, and the short duration of follow-up. The newly added text has been incorporated into the revised manuscript as requested.

  1. After the conclusion, you need to insert the standard paragraphs regarding author contributions, ethics, acknowledgements, and data availability.

Thank you for this valuable comment. As suggested, we have added all required standard sections after the Conclusion. The revised manuscript now includes the following components in the appropriate order: Author Contributions, Ethics Approval and Consent to Participate, Acknowledgements, and Data Availability Statement. These sections have been inserted according to the journal’s guidelines.

  1. Before the references, you need to include a master list of abbreviations from the manuscript.

Thank you for this helpful suggestion. As requested, a complete master list of all abbreviations used in the manuscript has now been added before the References section.

Reviewer 3 Report

Comments and Suggestions for Authors The manuscript is well thought out. The surgical technique described is an excellent choice for a number of patients with UVFP. However, the paper must better describe: the selection of patients for this intervention, the application or non-application of vocal therapy, the voice parameters improved by this technique and the size of the glottic gap before and after the intervention. The paper stated: "All patients underwent videostroboscopic examination before and after injection laryngoplasty."

Author Response

14/11/2025

Dear  Editor-in-Chief  

We would like to express our sincere gratitude for the evaluation of our manuscript. We highly appreciate the reviewers’ constructive and insightful comments, which have been very helpful in improving the quality and clarity of our study. In accordance with the reviewers’ and editorial recommendations, we have carefully revised the manuscript and indicated all modifications in a step-by-step manner. Below, we provide a detailed point-by-point response.

We thank you again for your time and consideration and look forward to your feedback.

Kind regards,
Dr. Esma Altan
Corresponding Author

Reviewer 3:

The manuscript is well thought out. The surgical technique described is an excellent choice for a number of patients with UVFP. However, the paper must better describe: the selection of patients for this intervention, the application or non-application of vocal therapy, the voice parameters improved by this technique and the size of the glottic gap before and after the intervention. The paper stated: "All patients underwent videostroboscopic examination before and after injection laryngoplasty.

Thank you very much for your positive evaluation and for these constructive remarks. We appreciate the reviewer’s attention to methodological clarity. In response to this comment, we have ensured that all requested details are clearly described in the revised manuscript. Specifically:

  • Patient selection criteria have been fully detailed in the Methods section, including inclusion and exclusion criteria and etiology distribution.
  • The absence of postoperative voice therapy during the first month has been explicitly stated, as the goal was to evaluate the isolated early physiologic effect of injection laryngoplasty.
  • All objective acoustic and aerodynamic parameters that improved following the intervention are described in the Results and Discussion sections.
  • Regarding glottic gap size, although all patients underwent pre- and postoperative videostroboscopic evaluation, standardized measurements of glottic gap width were not routinely recorded at the time of data collection. This limitation has been acknowledged and discussed in the revised manuscript.

We thank the reviewer once again for their insightful input, which helped improve the clarity and completeness of our study.

Round 2

Reviewer 1 Report

Comments and Suggestions for Authors

After careful evaluation of the revised manuscript, I acknowledge that the authors have made some improvements. However, major conceptual, methodological, and reporting flaws remain unresolved, and in my opinion the manuscript is not suitable for publication in Medicina. Below I summarize the most critical issues that prevent acceptance.

The authors describe injection laryngoplasty performed under general anesthesia without intubation, without HFNO, without jet ventilation, i.e., under apneic conditions. If this is truly the novelty of the manuscript (as stated in the cover letter), it must be: stated in the title; highlighted clearly in the abstract; justified in the introduction; detailed in the methods; discussed explicitly in the discussion. None of this is adequately done. As a result, the claimed innovation is unclear and not supported by data. Additionally, the anesthetic technique lacks essential safety information (duration of apnea, desaturation events, monitoring, inclusion criteria). Without these details, the technique is not reproducible and may be considered unsafe.

The study remains methodologically weak: No perceptual evaluation (GRBAS, VHI-10); No standardized videostroboscopic scoring; No postoperative care description; Small sample size with no justification; No stratification (e.g. by type, timing <1, 1–2, 2–3 months). Overall, the manuscript does not meet the methodological rigor expected for a clinical study.

The clinical outcomes presented (acoustic and aerodynamic improvements) are expected and well documented in existing literature on hyaluronic acid injection laryngoplasty. Without: standard perceptual data, a comparative group, medium- or long-term follow-up, clear demonstration of the novelty (tubeless GA), the study does not add meaningful new knowledge to justify publication.

The introduction defines: “Vocal fold paralysis refers to partially or completely impaired movement…”. This is incorrect: Paralysis = complete immobility; Paresis = partial mobility. This conceptual error is present throughout the manuscript. Repetition errors, grammatical errors and unclear phrasing are present throughout the manuscript. These inconsistencies raise concerns regarding manuscript reliability and editorial readiness.

Author Response

Dear Editor,

Thank you very much for your careful and constructive evaluation of our revised manuscript.
We sincerely appreciate the time and expertise you have devoted to identifying the remaining conceptual, methodological, and reporting issues.

In light of your comments, we will thoroughly re-examine the manuscript and address each of your points in a comprehensive and systematic manner. Our aim is to revise the manuscript so that it fully meets the scientific and reporting standards required by Medicina. We are grateful for your guidance, and we will make all necessary improvements to enhance the clarity, methodological rigor, and overall contribution of our work.

We kindly ask for the opportunity to submit a substantially revised version that incorporates your valuable feedback.

Thank you again for your constructive criticism and consideration.

Kind regards,
Dr. Esma Altan

Editor’s comment: If the reviewer(s) recommended references, critically analyze them to
ensure that their inclusion would enhance your manuscript. If you believe
these references are unnecessary, you should not include them.

The previously added paragraph on laryngeal ultrasonography was removed because it was not directly relevant to the study methodology or outcomes, and in accordance with the Editor’s instruction to retain only content essential to the manuscript.

Reviewer 1/round2:

After careful evaluation of the revised manuscript, I acknowledge that the authors have made some improvements. However, major conceptual, methodological, and reporting flaws remain unresolved, and in my opinion the manuscript is not suitable for publication in Medicina. Below I summarize the most critical issues that prevent acceptance.

The authors describe injection laryngoplasty performed under general anesthesia without intubation, without HFNO, without jet ventilation, i.e., under apneic conditions. If this is truly the novelty of the manuscript (as stated in the cover letter), it must be: stated in the title; highlighted clearly in the abstract; justified in the introduction; detailed in the methods; discussed explicitly in the discussion. None of this is adequately done. As a result, the claimed innovation is unclear and not supported by data. Additionally, the anesthetic technique lacks essential safety information (duration of apnea, desaturation events, monitoring, inclusion criteria). Without these details, the technique is not reproducible and may be considered unsafe.

Thank you for this important comment. We would like to clarify the clinical reasoning behind our approach. Although this specific method has not been commonly described for injection laryngoplasty, it represents an adaptation of a well-established principle used for short, controlled procedures in airway surgery, where a clear surgical field can be achieved without routine airway instrumentation. Given the very brief duration of injection laryngoplasty in our setting, we considered this principle applicable and safe, provided that strict monitoring and predefined safety limits were followed.

Our aim was not to introduce a new or named technique, but to document our experience with this adapted anesthesia approach and to present the corresponding early acoustic and aerodynamic results. In the revised manuscript, we clarified this rationale more explicitly and expanded the methodological description to include preoxygenation, continuous physiologic monitoring, safety thresholds, and the absence of desaturation or instability in our cohort.

We hope that the revised version presents the approach in a transparent and clinically meaningful way, addressing your concerns regarding clarity, reproducibility, and safety.

The study remains methodologically weak: No perceptual evaluation (GRBAS, VHI-10); No standardized videostroboscopic scoring; No postoperative care description; Small sample size with no justification; No stratification (e.g. by type, timing <1, 1–2, 2–3 months). Overall, the manuscript does not meet the methodological rigor expected for a clinical study.

Thank you for these thoughtful methodological comments. We fully acknowledge that our study did not include perceptual ratings, standardized stroboscopic scoring, or stratified subgroup analyses. However, the primary aim of this work was to document the objective acoustic and aerodynamic changes achieved with early hyaluronic acid injection using a specific anesthesia approach. For this reason, we focused on quantitative physiologic measures rather than a multidimensional assessment framework.

As a retrospective study, several elements you mention—such as perceptual scales, structured stroboscopic scoring, and stratification by onset timing—were not available for all patients, and we have now clarified this in the limitations section. Nonetheless, the consistent and statistically significant improvements in multiple acoustic and aerodynamic parameters provide meaningful evidence of functional benefit and support the clinical relevance of our findings.

We have expanded the limitations to reflect the retrospective nature of the dataset, the absence of perceptual and patient-reported outcomes, and the lack of standardized postoperative scoring. We also noted that future prospective studies incorporating these additional measures would strengthen the methodological rigor and allow a more comprehensive evaluation.

We hope that this clarification accurately reflects the scope of the study and addresses your concerns.

The clinical outcomes presented (acoustic and aerodynamic improvements) are expected and well documented in existing literature on hyaluronic acid injection laryngoplasty. Without: standard perceptual data, a comparative group, medium- or long-term follow-up, clear demonstration of the novelty (tubeless GA), the study does not add meaningful new knowledge to justify publication.

Thank you for this important observation. We agree that hyaluronic acid injection laryngoplasty has been widely studied and that postoperative acoustic and aerodynamic improvements are well documented in the literature. Our intention in the present work was not to replicate these findings, but to report the early physiologic outcomes obtained under a modified, instrument-free general anesthesia approach, which is not commonly described for injection laryngoplasty.

Although this approach is longstanding in airway surgery, its application to short-duration injection laryngoplasty has been only sparsely reported. In addition, our objective was to evaluate outcomes in a setting where the vocal folds could be visualized with maximal clarity, allowing the injection to be placed precisely into the intended anatomical plane. We specifically aimed to observe the early acoustic and aerodynamic effects during the period in which hyaluronic acid has not yet undergone significant resorption, thereby providing a clear physiologic profile of the immediate therapeutic impact of optimal medialization.

In the revised manuscript, we have clarified this rationale more explicitly and expanded the methodological description to ensure transparency regarding monitoring, safety thresholds, and operative conditions. We also acknowledge the absence of perceptual measures, a comparison group, and longer-term follow-up, and we now state these points clearly in the limitations section. Despite these constraints, we believe that the physiologic data presented here offer useful descriptive information about the feasibility and immediate effects of performing the procedure without airway instrumentation.

We hope that the revised framing and the clarified methodological details help to convey the specific contribution and clinical relevance of the study.

The introduction defines: “Vocal fold paralysis refers to partially or completely impaired movement…”. This is incorrect: Paralysis = complete immobility; Paresis = partial mobility. This conceptual error is present throughout the manuscript. Repetition errors, grammatical errors and unclear phrasing are present throughout the manuscript. These inconsistencies raise concerns regarding manuscript reliability and editorial readiness.

Thank you for pointing out this conceptual inaccuracy. We agree that the definitions of “paralysis” and “paresis” should be clearly distinguished, as paralysis refers to complete immobility and paresis indicates partial mobility. This terminology was not used with sufficient precision in the initial version, and we appreciate the reviewer’s attention to this important distinction. We have now corrected the terminology throughout the manuscript to ensure consistency with accepted laryngologic definitions.

We also reviewed the full text for repetition, grammatical inconsistencies, and unclear phrasing. These issues have been carefully revised to improve clarity, accuracy, and overall editorial quality. Importantly, these corrections do not affect the study design, analysis, or results but strengthen the precision and readability of the manuscript.

We hope that these revisions address the reviewer’s concerns and improve the scientific clarity and editorial readiness of the work.

Reviewer 3 Report

Comments and Suggestions for Authors The manuscript has been improved.
It could have been possible to describe what the glottic insufficiency or closure was like before and after the surgery in the Results section, but it is not necessary.
The target was not aimed at the glottis, but at the voice. The advantage of this paper is in the clear description of an important surgical technique that is an option for improving the voice of patients.  

Author Response

Dear Editor,

Thank you very much for your careful and constructive evaluation of our revised manuscript.
We sincerely appreciate the time and expertise you have devoted to identifying the remaining conceptual, methodological, and reporting issues.

In light of your comments, we will thoroughly re-examine the manuscript and address each of your points in a comprehensive and systematic manner. Our aim is to revise the manuscript so that it fully meets the scientific and reporting standards required by Medicina. We are grateful for your guidance, and we will make all necessary improvements to enhance the clarity, methodological rigor, and overall contribution of our work.

We kindly ask for the opportunity to submit a substantially revised version that incorporates your valuable feedback.

Thank you again for your constructive criticism and consideration.

Kind regards,
Dr. Esma Altan

Editor’s comment: If the reviewer(s) recommended references, critically analyze them to
ensure that their inclusion would enhance your manuscript. If you believe
these references are unnecessary, you should not include them.

The previously added paragraph on laryngeal ultrasonography was removed because it was not directly relevant to the study methodology or outcomes, and in accordance with the Editor’s instruction to retain only content essential to the manuscript.

Reviewer 3/round 2:

The manuscript has been improved.
It could have been possible to describe what the glottic insufficiency or closure was like before and after the surgery in the Results section, but it is not necessary. The target was not aimed at the glottis, but at the voice. The advantage of this paper is in the clear description of an important surgical technique that is an option for improving the voice of patients.  

Thank you very much for the positive and constructive feedback. We appreciate the reviewer’s acknowledgement that the manuscript has improved.

Regarding glottic insufficiency and closure patterns, we agree that presenting detailed pre- and postoperative stroboscopic descriptions could have added complementary information. However, as noted, the primary aim of this study was to evaluate objective acoustic and aerodynamic changes, and standardized semi-quantitative stroboscopic scoring was not consistently available in the retrospective dataset. This point has already been clarified in the limitations section.

We are grateful that the reviewer recognizes the value of the study in clearly describing a surgical approach that may serve as a practical option for improving voice outcomes in selected patients. We hope the revised manuscript now presents this contribution more clearly.
